∂ | **Open Peer Review** | Antimicrobial Chemotherapy | Research Article

# Antimicrobial resistance, virulence gene profiles, and molecular epidemiology of enterococcal isolates from patients with urinary tract infections in Shanghai, China

Weiyi Wu,[1] Shuzhen Xiao,[2,3] Lizhong Han,[2,3] Qiong Wu[4]

**ABSTRACT** Urinary tract infections (UTIs) are among the most prevalent infectious diseases, yet there is still limited understanding of the epidemiology of Enterococcal strains isolated from UTI patients in Shanghai. This study aims to elucidate the antimicrobial resistance profiles, virulence gene carriage, and molecular epidemiology of selected Enterococcal strains from UTI patients in Shanghai. A cohort of 80 *Enterococcus faecalis* and 40 *Enterococcus faecium* clinical isolates were randomly selected from UTI patients from October 2022 to March 2023. No vancomycin-resistant strains were identified based on minimum inhibitory concentration (MIC) testing. However, five strains of linezolid-resistant *E. faecalis* were identified, all of which were confirmed to be *optrA*-positive through whole-genome sequencing (WGS), with ST300 being reported as the first instance of this ST type in China. Polymerase chain reaction (PCR) assays were employed to ascertain the presence of virulence genes and multi-locus sequence type (MLST). In *E. faecalis*, the most common virulence genes were *asaI* (75%), *gelE* (65%), *esp* (52.5%), and *cylA* (47.5%). In contrast, *E. faecium* primarily exhibited *esp* (65%) and *hyl* (12.5%). Among the *E. faecalis* strains, 21 distinct MLST types were identified, with ST16 and ST179 prevailing. Conversely, *E. faecium* exhibited only five MLST types, with ST78 being predominant. The prevalence of *E. faecalis* CC16 and *E. faecium* CC17 further complicates the treatment landscape for Enterococcal UTIs.

**IMPORTANCE** This study highlighted the critical need to understand Enterococcal strains causing UTIs in Shanghai, given their high prevalence. By assessing antimicrobial resistance profiles, virulence gene presence, and molecular epidemiology, the research offered valuable insights into the local epidemiology of *Enterococcus faecalis* and *Enterococcus. faecium*. Identifying linezolid-resistant strains, all of which carry the *optrA* gene, including the first report of ST300 in China and recognizing dominant MLST types, such as ST16 and ST179 for *E. faecalis* and ST78 for *E. faecium*, are vital for guiding treatment and addressing the challenges these infections present. The data emphasize the need for ongoing surveillance and customized therapeutic approaches to combat emerging resistance and virulence factors in Enterococcal UTIs.

**KEYWORDS** urinary tract infections, *Enterococcus*, antimicrobial resistance, *optrA*, virulence genes, molecular epidemiology

Urinary tract infections (UTIs) are one of the most common infectious diseases worldwide, with an estimated 40% of women and 12% of men experiencing at least one symptomatic UTI in their lifetime (1). According to statistics from the CHINET surveillance network, *Enterococcus* species are established as the second most prevalent pathogen in UTIs. Furthermore, mortality rates among inpatients with UTIs attributed to *Enterococcus* species surpass those associated with other pathogens (2–4). The risk

Address correspondence to Lizhong Han, 13916291150@163.com, or Qiong Wu, joanwu29@hotmail.com.

The authors declare no conflict of interest.

*[This article was published on 29 November 2024 with an error in the first affiliation. The affiliation was corrected in the current version, posted on 6 December 2024.]*

factors associated with Enterococcal UTIs include male gender, urinary catheterization, prior antibiotic usage, urological malignancies, and various forms of immunosuppression (5). Enterococci contribute significantly to the incidence of cystitis, prostatitis, and epididymitis associated with UTIs (6).

Owing to intrinsic and acquired antibiotic resistances, tolerance to disinfectants (7), Enterococci at the genus level are the second most common pathogens associated with causing hospital-acquired infections (HAI) in both Europe and the United States (8). Due to the emergence and rapid spread of Vancomycin-resistant *Enterococcus* (VRE) in recent years, there has been a growing emphasis on the significance of alternative antimicrobials, including linezolid (9). In recent years, the widespread clinical use of linezolid has led to an increasing number of reports on linezolid-resistant Gram-positive pathogens, highlighting the growing risk of resistance transmission (10). Resistance to linezolid was attributed to mutations in the 23S rRNA and genes encoding the 50S ribosomal proteins L3 (*rplC*) and L4(*rplD*), the acquisition of genes, such as *optrA* and *poxtA*, encoding an ATP-binding cassette (ABC-F) protein, and *cfr* variants coding for 23S rRNA methyltransferase (11–13).

In addition to resistance mechanisms, the virulence factors of Enterococci play a significant role in their pathogenic potential. Virulence genes, such as gelatinase (*gelE*), hyaluronidase (*hyl*), aggregation substances (*asaI*), Enterococcal surface protein (*esp*), and cytolysin (*cylA*), are implicated in the processes including adhesion, biofilm formation, bacterial dissemination, aggregation, and facilitated conjugation (14–16). Consequently, UTIs stemming from *Enterococcus* species present a formidable global challenge.

This study endeavors to investigate the antibiotic susceptibility profiles, virulence factors, and MLST patterns of selected *Enterococcus* species associated in UTIs in Shanghai. By analyzing bacteriological data on Enterococcal isolates, an *optrA*-positive linezolid-resistant *E. faecalis* strain of ST300 was first detected in China, to our knowledge; this strain also carried the *fexA*, *ermA/B*, and *lsa(A)* genes.

## RESULTS

### Clinical characteristics

A total of 120 patients were enrolled in the study, including 63 males (52.5%) and 57 females (47.5%). The median age of these patients was 68 years, with a range of 18 to 102 years. Patients older than 60 years were more likely to develop Enterococcal UTIs. Among the 120 strains, 56.3% (45/80) of *E. faecalis* and 92.5% (37/40) of *E. faecium* were acquired though HAI. Furthermore, 96.7% (116/120) of the patients had polymicrobial infections, and catheter-associated UTIs accounted for 64.2% (77/120). Cystitis was observed in 89.2% (107/120) of the patients. Those treated with steroid, immunosuppressant, or antibiotic, were more likely to be infected by *E. faecium*. Patients with urological malignancy comorbidities are more prone to *E. faecium* infections, whereas those with pneumonia complications are more likely to be infected by *E. faecalis* (Table 1).

### Antimicrobial susceptibility testing

All of the 120 isolates were susceptible to vancomycin and teicoplanin. Five isolates of *E. faecalis* were resistant to linezolid. All of the *E. faecium* were resistant to ampicillin, whereas the resistance rate of *E. faecalis* was 6.2%. Additionally, *E. faecalis* demonstrated higher sensitivity to nitrofurantoin and fosfomycin compared to *E. faecium* ($P < 0.05$) (Table 2).

### Distribution of virulence genes

In *E. faecalis* isolates, *asaI* (75.0%) represented the predominant virulence factors, followed by *gelE* (65.0%), *esp* (52.5%), and *cylA* (47.5%). Conversely, *hyl* was not detected in any of the *E. faecalis* isolates. In contrast, *E. faecium* isolates predominantly carried *esp*

**TABLE 1** Comparison of the characteristics of patients infected with *E. faecalis* and *E. faecium*[a]

| Characteristics | *E. faecalis* (n = 80) | *E. faecium* (n = 40) | Total (n = 120) | *P*-value (Efa vs Efm) |
|---|---|---|---|---|
| Patient origin | | | | |
| Age | | | | |
| <60 | 31 | 6 | 37 | 0.008 |
| ≥60 | 49 | 34 | 83 | |
| Sex | | | | |
| Females | 37 | 20 | 57 | 0.698 |
| Males | 43 | 20 | 63 | |
| Source of UTIs | | | | |
| Community acquired | 35 | 3 | 38 | 0.000 |
| Hospital acquired | 45 | 37 | 82 | |
| Infection type | | | | |
| Polymicrobial | 79 | 37 | 116 | 0.072 |
| Monomicrobial | 1 | 3 | 4 | |
| Catheter-associated | 46 | 31 | 77 | 0.031 |
| Infection severity | | | | |
| Cystitis | 70 | 37 | 107 | 0.276 |
| Pyelonephritis | 2 | 2 | 4 | |
| Asymptomatic bacteriuria | 8 | 1 | 9 | |
| Drug usage | | | | |
| Steroid/Immunosuppressor | 13 | 17 | 30 | 0.002 |
| Antibiotics | 21 | 25 | 46 | 0.000 |
| Comorbidities | | | | |
| Urological malignancy | 14 | 0 | 14 | 0.005 |
| Urinary calculi | 10 | 1 | 11 | 0.074 |
| Pneumonia | 1 | 16 | 17 | 0.000 |

[a]Efa, *E. faecalis*; Efm, *E. faecium*.

(65.0%) and *hyl* (12.5%). Significant differences were observed in the prevalence of *gelE*, *asaI*, *cylA*, and *hyl* between *E. faecalis* and *E. faecium* (*P* < 0.05) (Table 2).

## MLST typing

Sequence types (STs), determined through multilocus sequence typing (MLST), were assigned to all isolates (Fig. 1). In *E. faecalis*, 21 distinct STs were identified, with ST16 being the most prevalent (*n* = 27, 33.8%), followed by ST179 (*n* = 15, 18.8%). BURST analysis revealed that ST16 and ST179 clustered within the same clonal complex (CC16), ST179 (5/1/1/3/7/1/6) representing a single locus variant (SLV) of ST16 (5/1/1/3/7/7/6). Among the linezolid-resistant *E. faecalis* isolates, two belonged to ST16, and one each to ST179, ST376, and ST300. The correlation between MLST phenotype, antimicrobial resistance, and virulence genes in *E. faecalis* UTI isolates is outlined in Table 3. The carrying rates of *gelE*, *esp* and *cylA* in CC16 *E. faecalis* were higher than those of other STs (Table 4). All *E. faecium* isolates were grouped into CC17, comprising five different STs (ST78, ST555, ST80, ST192, ST17), with ST78 being the predominant type (*n* = 26, 65.0%) (Table 5).

## Linezolid resistance mechanisms in *E. faecalis*

All five linezolid resistance *E. faecalis* strains harbored *optrA*. Comparison of the *OptrA* amino acid sequences of these five linezolid resistance strains with that of the original *optrA* from *E. faecalis* E349 (designated as the wild type) revealed that one strain exhibited three novel point mutations at positions 104 (Ile→Arg), 176 (Tyr→Asp), and 256 (Glu→Lys) (RDK variant). No amino acid mutations were detected in the 23S rRNA or in

**TABLE 2** Comparison of antimicrobial resistance patterns, virulence genes of *E. faecalis* and *E. faecium*[a]

| | *E. faecalis* (*n* = 80) | *E. faecium* (*n* = 40) | *P*-value |
|---|---|---|---|
| Antibiotics resistance, n (%) | | | |
| Ampicillin | 5 (6.2) | 40 (100) | <0.001 |
| High-level gentamicin | 28 (35) | 15 (37.5) | 0.788 |
| Fosfomycin | 3 (3.8) | 7 (17.5) | 0.010 |
| Nitrofurantoin | 2 (2.5) | 25 (62.5) | <0.001 |
| Linezolid | 5 (6.2) | 0 (0) | 0.258 |
| Vancomycin | 0 (0) | 0 (0) | - |
| Teicoplanin | 0 (0) | 0 (0) | - |
| Virulence genes, n (%) | | | |
| *gelE* | 52 (65) | 0 (0) | <0.001 |
| *hyl* | 0 (0) | 5 (12.5) | 0.006 |
| *asal* | 60 (75) | 0 (0) | <0.001 |
| *esp* | 42 (52.5) | 26 (65) | 0.193 |
| *cylA* | 38 (47.5) | 0 (0) | <0.001 |

[a]The "-" in the vancomycin and teicoplanin rows under the "*P*" columns means the *P*-value was not calculated since all strains were susceptible.

the 50S ribosomal proteins L3 (*rplC*) and L4(*rplD*). Additionally, the *cfr* and *poxtA* genes were not identified in any of the strains. All isolates contained acquired genes associated with resistance to oxazolidinones (*optrA*), phenicols (*fexA*), macrolides, and lincosamides (*ermA/B*, *lsaA*), which were present in all the isolates (Table 6).

## DISCUSSION

Enterococci, classified within the Gram-positive cocci, primarily attributed to *E. faecalis* and *E. faecium* , have been prevalent pathogens causing UTIs. *Enterococcus* species are responsible for a small fraction of UTIs occurring in the community, but it is also primarily associated with hospital-acquired UTIs (17). Hospital-acquired UTIs are associated with urinary catheterization, prior antibiotic usage, and various forms of immunosuppression (5). In our study, 70.8% patients were HAI, among which 93.9% were catheter-associated infections. Catheters are an idea setting for bacterial growth as they provide a surface for biofilm adhesion and disrupt the bladder environment. The patient' sex is an important consideration when treating UTIs. Previous studies (5, 18, 19) have indicated a higher prevalence of Enterococci-associated UTIs in males, particularly in the context of complex

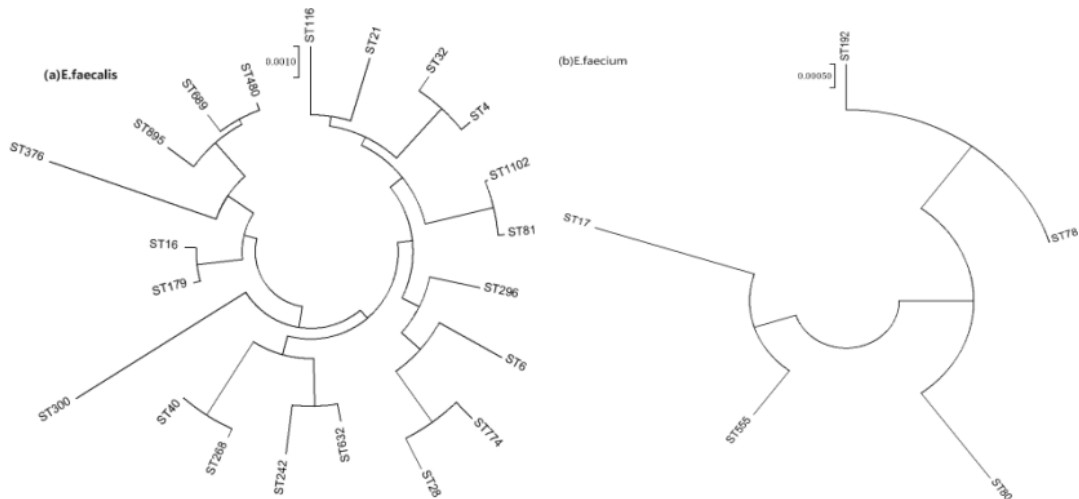

**FIG 1** Phylogenetic tree of *E. faecalis* (a) and *E. faecium* (b) isolated from UTI patients based on seven conserved housekeeping genes.

**TABLE 3** Relationship of MLST phenotype with antimicrobial resistance and virulence genes in *E. faecalis*[a]

| MLST | NO | % | AMP | HLGN | NIT | FOS | LNZ | TEC | VAN | gelE | hyl | asal | esp | cylA |
|---|---|---|---|---|---|---|---|---|---|---|---|---|---|---|
| ST16 | 27 | 33.75 | 1 (3.7%) | 12 (44.4%) | 1 (3.7%) | 1 (3.7%) | 2 (7.4%) | 0 (0.0%) | 0 (0.0%) | 4 (14.8%) | 0 (0.0%) | 18 (66.7%) | 18 (66.7%) | 17 (63.0%) |
| ST179 | 15 | 18.75 | 0 (0.0%) | 4 (26.7%) | 1 (6.7%) | 1 (6.7%) | 1 (6.7%) | 0 (0.0%) | 0 (0.0%) | 15 (100.0%) | 0 (0.0%) | 15 (100.0%) | 11 (73.3%) | 10 (66.7%) |
| ST4 | 8 | 10 | 4 (50.0%) | 3 (37.5%) | 0 (0.0%) | 0 (0.0%) | 0 (0.0%) | 0 (0.0%) | 0 (0.0%) | 8 (100.0%) | 0 (0.0%) | 7 (87.5%) | 5 (62.5%) | 5 (62.5%) |
| ST774 | 6 | 7.5 | 4 (66.7%) | 0 (0.0%) | 0 (0.0%) | 0 (0.0%) | 0 (0.0%) | 0 (0.0%) | 0 (0.0%) | 6 (100.0%) | 0 (0.0%) | 5 (83.3%) | 2 (33.3%) | 2 (33.3%) |
| ST81 | 3 | 3.75 | 0 (0.0%) | 0 (0.0%) | 0 (0.0%) | 0 (0.0%) | 0 (0.0%) | 0 (0.0%) | 0 (0.0%) | 3 (100.0%) | 0 (0.0%) | 1 (33.3%) | 0 (0.0%) | 1 (33.3%) |
| ST895 | 3 | 3.75 | 0 (0.0%) | 1 (33.3%) | 0 (0.0%) | 0 (0.0%) | 0 (0.0%) | 0 (0.0%) | 0 (0.0%) | 0 (0.0%) | 0 (0.0%) | 3 (100.0%) | 3 (100.0%) | 1 (33.3%) |
| ST6 | 2 | 2.5 | 0 (0.0%) | 2 (100.0%) | 0 (0.0%) | 0 (0.0%) | 0 (0.0%) | 0 (0.0%) | 0 (0.0%) | 2 (100.0%) | 0 (0.0%) | 2 (100.0%) | 0 (0.0%) | 1 (50.0%) |
| ST268 | 2 | 2.5 | 0 (0.0%) | 0 (0.0%) | 0 (0.0%) | 1 (50.0%) | 0 (0.0%) | 0 (0.0%) | 0 (0.0%) | 2 (100.0%) | 0 (0.0%) | 0 (0.0%) | 0 (0.0%) | 0 (0.0%) |
| ST28 | 2 | 2.5 | 0 (0.0%) | 1 (50.0%) | 0 (0.0%) | 0 (0.0%) | 0 (0.0%) | 0 (0.0%) | 0 (0.0%) | 2 (100.0%) | 0 (0.0%) | 2 (100.0%) | 0 (0.0%) | 2 (100.0%) |
| ST1102 | 1 | 1.25 | 0 (0.0%) | 0 (0.0%) | 0 (0.0%) | 0 (0.0%) | 0 (0.0%) | 0 (0.0%) | 0 (0.0%) | 1 (100.0%) | 0 (0.0%) | 1 (100.0%) | 0 (0.0%) | 0 (0.0%) |
| ST116 | 1 | 1.25 | 0 (0.0%) | 1 (100.0%) | 0 (0.0%) | 0 (0.0%) | 0 (0.0%) | 0 (0.0%) | 0 (0.0%) | 1 (100.0%) | 0 (0.0%) | 1 (100.0%) | 0 (0.0%) | 0 (0.0%) |
| ST21 | 1 | 1.25 | 0 (0.0%) | 0 (0.0%) | 0 (0.0%) | 0 (0.0%) | 0 (0.0%) | 0 (0.0%) | 0 (0.0%) | 1 (100.0%) | 0 (0.0%) | 0 (0.0%) | 0 (0.0%) | 0 (0.0%) |
| ST242 | 1 | 1.25 | 0 (0.0%) | 0 (0.0%) | 0 (0.0%) | 0 (0.0%) | 0 (0.0%) | 0 (0.0%) | 0 (0.0%) | 1 (100.0%) | 0 (0.0%) | 0 (0.0%) | 0 (0.0%) | 0 (0.0%) |
| ST296 | 1 | 1.25 | 0 (0.0%) | 0 (0.0%) | 0 (0.0%) | 0 (0.0%) | 0 (0.0%) | 0 (0.0%) | 0 (0.0%) | 1 (100.0%) | 0 (0.0%) | 1 (100.0%) | 0 (0.0%) | 0 (0.0%) |
| ST300 | 1 | 1.25 | 0 (0.0%) | 0 (0.0%) | 0 (0.0%) | 0 (0.0%) | 1 (100.0%) | 0 (0.0%) | 0 (0.0%) | 1 (100.0%) | 0 (0.0%) | 1 (100.0%) | 0 (0.0%) | 0 (0.0%) |
| ST32 | 1 | 1.25 | 0 (0.0%) | 0 (0.0%) | 0 (0.0%) | 0 (0.0%) | 0 (0.0%) | 0 (0.0%) | 0 (0.0%) | 1 (100.0%) | 0 (0.0%) | 1 (100.0%) | 0 (0.0%) | 0 (0.0%) |
| ST376 | 1 | 1.25 | 0 (0.0%) | 0 (0.0%) | 0 (0.0%) | 0 (0.0%) | 1 (100.0%) | 0 (0.0%) | 0 (0.0%) | 1 (100.0%) | 0 (0.0%) | 1 (100.0%) | 0 (0.0%) | 0 (0.0%) |
| ST40 | 1 | 1.25 | 0 (0.0%) | 0 (0.0%) | 0 (0.0%) | 0 (0.0%) | 0 (0.0%) | 0 (0.0%) | 0 (0.0%) | 1 (100.0%) | 0 (0.0%) | 0 (0.0%) | 1 (100.0%) | 0 (0.0%) |
| ST480 | 1 | 1.25 | 0 (0.0%) | 1 (100.0%) | 0 (0.0%) | 0 (0.0%) | 0 (0.0%) | 0 (0.0%) | 0 (0.0%) | 0 (0.0%) | 0 (0.0%) | 0 (0.0%) | 1 (100.0%) | 0 (0.0%) |
| ST632 | 1 | 1.25 | 0 (0.0%) | 0 (0.0%) | 0 (0.0%) | 0 (0.0%) | 0 (0.0%) | 0 (0.0%) | 0 (0.0%) | 1 (100.0%) | 0 (0.0%) | 1 (100.0%) | 0 (0.0%) | 0 (0.0%) |
| ST689 | 1 | 1.25 | 0 (0.0%) | 0 (0.0%) | 0 (0.0%) | 0 (0.0%) | 0 (0.0%) | 0 (0.0%) | 0 (0.0%) | 0 (0.0%) | 0 (0.0%) | 1 (100.0%) | 0 (0.0%) | 0 (0.0%) |

[a]AMP, ampicillin; HLGN, high-level gentamicin; NIT, nitrofurantoin; FOS, fosfomycin; LNZ, linezolid; TEC, teicoplanin; VAN, vancomycin.

**TABLE 4** Comparison of antimicrobial resistance, virulence genes among CC16 and other STs in *E. faecalis*

| % | ST16 + ST179 (CC16) (*n* = 42) | Other STs (*n* = 38) | *P*-value |
|---|---|---|---|
| Ampicillin | 2.4 | 10.5 | 0.133 |
| High-level gentamicin | 38.1 | 31.6 | 0.542 |
| Nitrofurantoin | 4.8 | 0.0 | 0.173 |
| Fosfomycin | 4.8 | 2.6 | 0.616 |
| Linezolid | 7.1 | 5.3 | 0.729 |
| *gelE* | 45.2 | 86.8 | 0.000 |
| *asal* | 78.6 | 71.1 | 0.438 |
| *esp* | 69.1 | 34.2 | 0.002 |
| *cylA* | 64.3 | 29.0 | 0.002 |

UTIs. Consistent with these findings, our study highlights a propensity for higher rates of Enterococcal UTIs in males compared to females. The greater susceptibility of men to *Enterococcus* species infections compared to women is not well understood, though various hypotheses exist. One such hypothesis is that the prostate in men could harbor bacteria and develop micro-abscesses. Bacteria may then translocate from the intestines and colonize the prostate tissue (20).

The resistance and tolerance of *Enterococcus* species to a broad range of antibiotics present significant challenges in treatment. This adaptability is partly due to their highly flexible genetic structure, which facilitates the acquisition of mobile genetic elements, genome hybridization with other Enterococci, and gene transfer between species (17). Glycopeptide and oxazolidinone antibiotics are considered the last line of defense against resistant Enterococci. Moreover, ampicillin also remains an effective treatment for UTIs caused by resistant Enterococci when administered at high concentrations in the urine, even for the treatment of complicated UTI caused by VRE (21). In our study, the resistance rates to ampicillin among *E. faecium* and *E. faecalis* strains were 100% and 6.2%, respectively. The detection rate of VRE strains was lower than those reported in many international studies (22–24). However, five linezolid-resistant *E. faecalis* strains were identified in this study, with a resistance rate of 6.3%, which exceeds the rates reported in Spain (0.7%) (25), Austria (0.2%) (26), and Japan (0.2%) (27). Resistance to linezolid was verified attribute to mutations in the 23S rRNA and genes encoding the 50S ribosomal proteins L3 and L4 (*rplC* and *rplD*) or the acquisition of genes, such as *optrA,* encoding an ATP-binding cassette (ABC-F) protein, *cfr* and its variants coding for 23S rRNA methyltransferase (11), and *poxtA* associating with its mobility (28). All the strains carried the *optrA*, consistent with the findings reported by Wang et al. (29). Additionally, *optrA* encompasses several variants that demonstrate differing levels of resistance to linezolid (30). In contrast to the prevalent EDM variant found in Argentina (31), only one RDK variant was identified in our study, which is the most common *optrA* variant in China (29). The remaining four isolates were of the wild type. Unexpectedly, both the wild-type *optrA* gene and the RDK variant exhibited relatively high levels of resistance to linezolid compared to other variants (30). This study suggests that *optrA*-positive linezolid-resistant *E. faecalis* may not only exhibit resistance to phenicols (mediated by *fexA*) but also to streptogramins, lincosamides [mediated by *lsa(A)*], and macrolides (mediated by *erm*), highlighting the necessity for ongoing vigilance.

**TABLE 5** Relationship of MLST phenotype with antimicrobial resistance and virulence genes in *E. faecium*[a]

| MLST | NO | % | AMP | HLGN | NIT | FOS | LNZ | TEC | VAN | *gelE* | *hyl* | *asal* | *esp* | *cylA* |
|---|---|---|---|---|---|---|---|---|---|---|---|---|---|---|
| ST78 | 26 | 65 | 26 (100.0%) | 9 (34.6%) | 16 (61.6%) | 5 (19.2%) | 0 (0.0%) | 0 (0.0%) | 0 (0.0%) | 0 (0.0%) | 3 (11.5%) | 0 (0.0%) | 16 (61.5%) | 0 (0.0%) |
| ST555 | 7 | 17.5 | 7 (100.0%) | 2 (28.6%) | 4 (57.1%) | 0 (0.0%) | 0 (0.0%) | 0 (0.0%) | 0 (0.0%) | 0 (0.0%) | 2 (28.6%) | 0 (0.0%) | 6 (85.7%) | 0 (0.0%) |
| ST80 | 4 | 10 | 4 (100.0%) | 2 (50.0%) | 4 (100.0%) | 1 (25.0%) | 0 (0.0%) | 0 (0.0%) | 0 (0.0%) | 0 (0.0%) | 0 (0.0%) | 0 (0.0%) | 2 (50.0%) | 0 (0.0%) |
| ST192 | 2 | 5 | 2 (100.0%) | 1 (50.0%) | 1 (50.0%) | 0 (0.0%) | 0 (0.0%) | 0 (0.0%) | 0 (0.0%) | 0 (0.0%) | 0 (0.0%) | 0 (0.0%) | 2 (100.0%) | 0 (0.0%) |
| ST17 | 1 | 2.5 | 1 (100.0%) | 0 (0.0%) | 0 (0.0%) | 1 (100.0%) | 0 (0.0%) | 0 (0.0%) | 0 (0.0%) | 0 (0.0%) | 0 (0.0%) | 0 (0.0%) | 0 (0.0%) | 0 (0.0%) |

[a]AMP, ampicillin; HLGN, high-level gentamicin; NIT, nitrofurantoin; FOS. fosfomycin; LNZ, linezolid; TEC, teicoplanin; VAN, vancomycin.

**TABLE 6** Linezolid resistance mechanisms and associated genes in *E. faecalis*[a,b]

| Strain no. | MLST | *oprtA* | Fenxing | *cfr* | *cfr(B)* | *poxtA* | Mutation in 23srRna | | Mutation in *rplC/D* | *fexA* | *ermA* | *ermB* | *lsa(A)* |
|---|---|---|---|---|---|---|---|---|---|---|---|---|---|
| | | | | | | | G2576U | G2505A | | | | | |
| NL1422 | ST179 | + | E349 | - | - | - | - | - | - | + | + | - | + |
| NL1548 | ST16 | + | E349 | - | - | - | - | - | - | + | + | - | + |
| NL1801 | ST16 | + | E349 | - | - | - | - | - | - | + | + | + | + |
| NL1817 | ST300 | + | E349 | - | - | - | - | - | - | + | + | + | + |
| NL1830 | ST376 | + | RDK | - | - | - | - | - | - | + | - | + | + |

[a]"-" indicates not detected.
[b]"+" indicates detected.

While antibiotic resistance remains a significant concern in Enterococcal infections, understanding the role of virulence genes is also essential, as these factors further complicate clinical outcomes. In our study, the prevalence of *esp* gene, which is crucial for urethral colonization (32), was found to exceed 50% in both *E. faecalis* and *E. faecium*. Consistent with previous research (33), the *gelE*, *cylA*, and *esp* genes were among the most common virulence factors identified in *E. faecalis* from patients with UTIs. Conversely, the *hyl* was detected only in *E. faecium* and is known to facilitate bacterial spread within tissues (34). Similar to previous research (27, 35), our results demonstrated that *E. faecalis* exhibited higher virulence compared to *E. faecium*, despite showing lower levels of both intrinsic and acquired antimicrobial resistance (8).

The *E. faecalis* isolates analyzed in this study were categorized into various STs, with ST16 (33.8%) and ST179 (18.8%) (CC16) being the most prevalent. These findings are consistent with reports from Japan (27) and suggest a possible clonal spread of *E. faecalis* causing UTIs in Shanghai. A 2018 report had speculated that ST4 *E. faecalis* might become the predominant strain in clinical infections across Asia (36). However, in our study, ST4 *E. faecalis* was identified in only 10% of cases. Among linezolid-resistant *E. faecalis*, ST16 was the most prevalent type, which aligns with a previous article (10). Notably, to the best of our knowledge, an *optrA*-positive linezolid-resistant *E. faecalis* strain of ST300 was first detected in China in this study; this strain also carried the *fexA*, *ermA/B*, and *lsa(A)* genes.

In the analysis of 40 of *E. faecium* strains, we identified five distinct STs, all classified within the CC17 clonal complex. ST78 was the predominant type, accounting for 65%, which closely resembled that in Algeria (37). The emergence of the ST80 VREFM strain dates back to its initial detection in Israel in 1997. Hammerum et al. (38) reported a significant increase in ST80 VREFM from 2005 to 2015 in Denmark, which aligns with research conducted in India (39). However, in our study, the four ST80 *E. faecium* stains were not resistant to vancomycin. Given the significant connection between CC17 and hospital-acquired Enterococcal infections, there is a compelling need for heightened control measures.

In our study, *E. faecalis* exhibits a higher prevalence of virulence genes compared to *E. faecium*. MLST analysis reveals dominant ST16 and ST179 strains among *E. faecalis*, whereas *E. faecium* is predominantly associated with ST78. Notably, all five linezolid-resistant *E. faecalis* strains carried the *optrA*, with ST300 being the first report of this ST type in China. A primary limitation of this study is the relatively short duration of data collection. Additionally, the study is geographically limited to two hospitals in a specific region, which may not fully represent clinical practices across different regions. Furthermore, the relatively small sample size affects the generalizability of the findings and may limit the detection of less common phenomena. With the escalation of virulence genes and resistance in Enterococci implicated in UTIs, there is a pressing need for intensified frequency of assessments and research endeavors on this microbial cohort. A thorough understanding of the resistance mechanisms, carriage of virulence genes, and prevalent MLST strains among Enterococci is essential for predicting sudden shifts, establishing foundational frameworks for treating clinical UTIs caused by Enterococci, and effectively managing these infections.

## MATERIALS AND METHODS

### Bacterial strains

*E. faecalis* and *E. faecium* isolated from midstream urine cultures in clinical samples from Ruijin Hospital affiliated Shanghai Jiaotong University School of Medicine and the Sixth People's Hospital affiliated to Shanghai Jiaotong University School of Medicine from October 2022 to March 2023 were collected. A total of 120 strains were randomly selected, including 80 *E. faecalis* and 40 *E. faecium*, with the RAND() function in Microsoft Excel used to ensure an unbiased selection process. The VITEK MS Microbial Identification System (BioMérieux, France) was employed for species identification. This retrospective study was approved by the Ethics Committee of Shanghai Sixth People's Hospital (No. 2024-KY-051K).

### Antimicrobial susceptibility testing

The susceptibility of Enterococcal strains to common antimicrobial agents was determined using the disc diffusion method (Kirby-Bauer [K-B]) and the AST GP67 VITEK-2 system (BioMérieux, France) according to CLSI guidelines (2023). The antibiotics tested include ampicillin, high-level gentamicin, nitrofurantoin, linezolid, vancomycin, teicoplanin, and fosfomycin (Hangzhou Binhe Microbial Reagents Co.). *E. faecalis* ATCC 29212 and *S. aureus* ATCC 25923 were used as reference strains for antibiotic susceptibility testing. E-tests were performed to confirm the susceptibility of strains showing reduced sensitivity to linezolid.

### Resistance mechanism to linezolid determined by WGS

Five linezolid-resistant *E. faecalis* strains were submitted for WGS analysis. All gene models were then subjected to blastp against the non-redundant (NR) database in NCBI, as well as CARD (https://card.mcmaster.ca/), and SignalP (http://www.cbs.dtu.dk/services/SignalP/). LRE-Finder (version 1.0.0) (40) was applied to detect linezolid resistance genes [*optrA*, *cfr*, *cfr(B)*, and *poxtA*] and common mutations in the V domain of the 23S rRNA (G2576U or G2505A) in Enterococci. The *optrA* protein variants were identified by aligning to the complete *optrA* gene sequence from plasmid pE349 (GenBank Accession No. NG_048023.1) as a reference. To identify amino acid mutations in the *rplC* (ribosomal protein L3) and *rplD* (ribosomal protein L4) genes, the sequences were aligned and blasted against the wild-type sequences from *E. faecalis* ATCC 29212 (GenBank Accession No. CP008816.1).

### Detection of virulence factors

The bacterial deoxyribonucleic acid (DNA) was extracted by boiling method. PCR was used to detect the virulence genes *cylA*, *asaI*, *gelE*, *esp*, and *hyl* (41). All PCR fragments were sequenced, and the gene types were identified by comparing them to sequences in GenBank (https://blast.ncbi.nlm.nih.gov/Blast.cgi).

### MLST typing

Following the reference method on PubMLST (https://pubmlst.org/) (42, 43), MLST was conducted based on the Enterococci MLST database, and different numbers were assigned to each sequence of the seven housekeeping genes following the order. The sequences were then compared with known alleles of Enterococci in the MLST database to get the corresponding ST typing. The primers for the housekeeping genes used in *E. faecalis* are listed in Table S1, and those for *E. faecium* are provided in Table S2.

### Statistical analysis

The χ test was used for comparison. All tests were two-sided, and *P* < 0.05 was considered statistically significant. All data were analyzed using SPSS 26 statistical software.

## ACKNOWLEDGMENTS

This research received no specific grant from any funding agency in the public, commercial, or not-for-profit sectors.

## AUTHOR AFFILIATIONS

[1]Department of Laboratory Medicine, Huangpu Branch, Shanghai Ninth People's Hospital, Shanghai Jiao Tong University School of Medicine, Shanghai, China
[2]Department of Laboratory Medicine, Ruijin Hospital, Shanghai Jiao Tong University School of Medicine, Shanghai, China
[3]Department of Clinical Microbiology, Ruijin Hospital, Shanghai Jiao Tong University School of Medicine, Shanghai, China
[4]Department of Laboratory Medicine, Shanghai Sixth People's Hospital Affiliated to Shanghai Jiao Tong University School of Medicine, Shanghai, China

## AUTHOR ORCIDs

Qiong Wu http://orcid.org/0009-0002-8674-1370

## AUTHOR CONTRIBUTIONS

Weiyi Wu, Data curation, Writing – original draft | Shuzhen Xiao, Methodology | Lizhong Han, Conceptualization, Resources | Qiong Wu, Writing – review and editing

## ADDITIONAL FILES

The following material is available online.

### Supplemental Material

**Table S1 (Spectrum01217-24-s0001.docx).** The primers used for MLST typing of housekeeping genes in *Enterococcus faecalis*.
**Table S2 (Spectrum01217-24-s0002.docx).** The primers used for MLST typing of housekeeping genes in *Enterococcus faecium*.

### Open Peer Review

**PEER REVIEW HISTORY (review-history.pdf).** An accounting of the reviewer comments and feedback.

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
