## [Reviewer comments · Microbiology Spectrum]

Microbiology Spectrum

Antimicrobial Resistance, Virulence Gene Profiles, and Molecular Epidemiology of Enterococcal Isolates from Patients with Urinary Tract Infections in Shanghai, China

Weiye Wu, Shuzhen Xiao, Lizhong Han, and Qiong Wu

Corresponding Author(s): Qiong Wu, Shanghai Sixth People's Hospital Affiliated to Shanghai Jiao Tong University School of Medicine

Review Timeline:

Submission Date:	May 31, 2024
Editorial Decision:	July 19, 2024
Revision Received:	September 9, 2024
Editorial Decision:	October 21, 2024
Revision Received:	October 24, 2024
Editorial Decision:	October 24, 2024
Revision Received:	October 26, 2024
Accepted:	October 28, 2024

Editor: Tomefa Asempa

Reviewer(s): The reviewers have opted to remain anonymous.

Transaction Report:

DOI: <https://doi.org/10.1128/spectrum.01217-24>

Re: Spectrum01217-24 (Antimicrobial Resistance, Virulence Gene Profiles, and Molecular Epidemiology of Enterococcal Isolates from Patients with Urinary Tract Infections in Shanghai, China)

Dear Ms. Qiong Wu:

Thank you for the privilege of reviewing your work. Below you will find my comments, instructions from the Spectrum editorial office, and the reviewer comments.

Manuscript is currently wordy, encourage authors to limit manuscript to only pertinent info and significantly reduce words count in introduction and discussion. Provide citations for this sentence in the introduction: "Owing to inherent resistance, acquired resistance mechanisms, and tolerance to disinfectants, Enterococcus have emerged as the second most common pathogen causing hospital-acquired infections."

Revision Guidelines

Sincerely,
Tomefa Asempa
Editor
Microbiology Spectrum

Reviewer #1 (Comments for the Author):

This is an interesting study on Enterococcal strains isolated from UTI patients, although there are several aspects to clarify.

- 1, Introduction is dense with information, could it be simplified?
- 2, The Enterococcus clinical isolates were randomly selected from UTI patients, but the method was not mentioned in the text.
- 3, Table 4 , Relationship of MLST phenotype with E. faecalis, antimicrobial resistance and virulence gene , Do you mean "Relationship of MLST phenotype with antimicrobial resistance and virulence gene in E. faecalis" ?
- 4, Table 4 , It will be better to add proportion in brackets for antimicrobial resistance and virulence genes. For example, ST16, AMP,1 (3.7%)
- 5, Table 5, suggestions are similar to those for Table 4.
- 6, Line 237 , ST4 E. faecalis, as there also exist E. faecium
- 7,What is the mechanism of five E. faecalis isolates which were not susceptible to Linezolid? Whole genome sequencing is suggested to perform to get some clues.

Reviewer #2 (Comments for the Author):

The study by Weiyi Wu et al investigated the antimicrobial resistance profiles, virulence gene carriage, and molecular epidemiology of Enterococcal strains derived from UTI patients in Shanghai by disk diffusion method and PCR assays. The authors conclude that E. faecalis exhibits a higher prevalence of virulence genes compared to E. faecium. First of all, the content of the manuscript seems to be not capable of the journal of Microbiology Spectrum. Other comments about the manuscript are shown below.

1. For the Antimicrobial Susceptibility Testing, disc diffusion method was used. The MICs of antimicrobials agents were not tested. For Enterococcus spp., fosfomycin is an important antibiotics and the study is not included. The antimicrobial susceptibility of teicoplanin was shown in the Results but not in the Method.
2. For the Detection of Virulence Factors, the positive amplified products should be sequenced to further confirm.
3. For the MLST typing, the primers of the housekeeping genes used in the manuscript should be showed at least in the supplementary materials.
4. The part of introduction is too rigmarole.
5. The authors should distinguish the genes and proteins. For example, esp is gene and Esp is protein. The gene should be italic.
6. The manuscript is only a description draft and no mechanisms are studied. Too many tables are included, most of which are not meaningful.

Reviewer #3 (Comments for the Author):

This study investigated the antibiotic susceptibility, virulence factors, and multilocus sequence typing (MLST) patterns of urinary tract infection with enterococci in a hospital in Shanghai,China.

Comments:

- 1.In the manuscript, the authors state that this study involves patient enrollment. According to current ethical requirements, such research usually requires the approval of an ethics committee. However, there is no information on whether or not ethical approval has been obtained in the manuscript.
- 2.The methods and judgment criteria for antimicrobial susceptibility testing should be listed. According to CLSI's routine recommendation, the quality control strain for KB method is Staphylococcus aureus ATCC25923, rather than Enterococcus faecalis ATCC29212.

Reviewer #4 (Comments for the Author):

Please see attachment.

The study by Weiyi Wu et al investigated the antimicrobial resistance profiles, virulence gene carriage, and molecular epidemiology of Enterococcal strains derived from UTI patients in Shanghai by disk diffusion method and PCR assays. The authors conclude that *E. faecalis* exhibits a higher prevalence of virulence genes compared to *E. faecium*. First of all, the content of the manuscript seems to be not capable of the journal of Microbiology Spectrum. Other comments about the manuscript are shown below.

1. For the Antimicrobial Susceptibility Testing, disc diffusion method was used. The MICs of antimicrobials agents were not tested. For *Enterococcus spp.*, fosfomycin is an important antibiotics and the study is not included. The antimicrobial susceptibility of teicoplanin was shown in the Results but not in the Method.
2. For the Detection of Virulence Factors, the positive amplified products should be sequenced to further confirm.
3. For the MLST typing, the primers of the housekeeping genes used in the manuscript should be showed at least in the supplementary materials.
4. The part of introduction is too rigmarole.
5. The authors should distinguish the genes and proteins. For example, *esp* is gene and Esp is protein. The gene should be italic.
6. The manuscript is only a description draft and no mechanisms are studied. Too many tables are included, most of which are meaningful.

June 28, 2024

Manuscript ID: Spectrum01217-24

Title: Antimicrobial Resistance, Virulence Gene Profiles, and Molecular Epidemiology of Enterococcal Isolates from Patients with Urinary Tract Infections in Shanghai, China

Author(s): Weiyi Wu, Shuzhen Xiao, Lizhong Han, Qiong Wu

Referee Comments:

The authors perform a retrospective, multi-center, epidemiological, cohort study in Shanghai, China which included 120 patients with Enterococcal urinary tract infection (UTI) to determine antibiotic susceptibility profiles, virulence factors, and multilocus sequence typing (MLST) patterns of these isolates. The main difficulty in this paper is that *Enterococcus* species from a urinary source have previously been defined to be highly ampicillin susceptible with evidence of using this agent even in resistant strains given superior urinary concentrations. In this sense, there is less novelty to this retrospective study with a small sample size. More clinical details or outcomes combined with a larger focus on linezolid resistant *Enterococcus faecalis* strains would have more impact. I have the following comments:

General suggestions

- Include more clinical details and endpoints to baseline characteristics.
 - Were UTIs polymicrobial or monomicrobial?
 - Infection severity/type – cystitis, pyelonephritis, catheter-associated UTI?
 - Community or hospital-acquired infection?
- Include additional baseline characteristics regarding risk factors associated with Enterococcal UTIs as alluded to in lines 54-56.
 - Malignancy?
 - Steroid use or other immunosuppressing medications/conditions?
 - Prior antimicrobial use?
- Address study limitations in the discussion.
- Discussion in general is lengthy and lacks logical flow; would suggest refining (e.g., consider moving gene descriptions to the introduction)

Detailed suggestions

- Line 19: add “select” in front of Enterococcal strains since only species *faecalis* and *faecium* were observed; Enterococcal does not need to be italicized
- Line 20: spell out numbers (e.g., eighty rather than 80)
- Line 23: define PCR before utilizing abbreviation
- Lines 40-41: sentence tense changes; suggest separating into two separate sentences
- Line 41: missing reference
- Line 43: restructure sentence to, “... 20.7% of pathogens were isolated from urine cultures...”
- Line 47: UTI was already previously defined in line 37
- Line 52: Enterococci emerge
- Lines 56-57: Enterococci contribute
- Lines 65, 67, 69, 71, 74: add “species” following *Enterococcus*
- Line 70: missing reference
- Lines 73-74: add “select” in front of *Enterococcus*

- Line 75: Enterococci does not need to be italicized; since all UTIs in this cohort were not hospital-acquired, it may not be appropriate to conclude that the findings of this study will inform management of nosocomial infections; you may consider a subgroup analysis describing antimicrobial resistance patterns in patients from a hospitalized origin versus patients from a community origin to see if there are any notable differences to highlight
- Line 81: remove “s” from UTIs
- Lines 81-82: spell out “years” rather than using “y”
- Lines 83-84: remain consistent with percentage rounding to nearest tenth as in above lines
- Table 1
 - Define Efa and Efm as footnote below table or next to the respective species
 - Consider adding other baseline characteristics that place patients at increased risk for Enterococcal UTIs as alluded to in lines 54-57 (e.g., severity of illness, previous antimicrobial use, malignancy, immunosuppressive medications, UTI type)
- Line 90: linezolid does not need to be capitalized
- Lines 90-93: consider removing the statement and data for erythromycin; due to increasing in vivo resistance and poor urinary penetration of macrolide antimicrobials (excreted 6-14% unchanged in the urine), they are not reliable options clinically for the treatment of Enterococcal UTI
- Lines 100-102: correct “tigecycline” to “teicoplanin”; remove references to Chi-square value since this statistic was not displayed in table 2
- Lines 130-131: remain consistent with percentage rounding to nearest tenth as in above lines
- Line 135: remain consistent with order of data presented (n, %)
- Line 143: remove “s” from UTIs
- Table 4: define the antimicrobial abbreviations used as a footnote (e.g., AMP = ampicillin)
- Line 164: Enterococci does not need to be italicized
- Line 166: missing reference
- Line 167: UTIs already defined; missing reference
- Line 171: capitalize Enterococci
- Line 175: capitalize Enterococci; urinary tract infection already defined
- Lines 176-177: sentence does not make grammatical sense; consider restructuring as follows: “UTIs affect diverse populations with a higher incidence rate among females, ranging from 62.6% to 86.9%.”
- Line 178: capitalize Enterococci
- Line 179: remove “s” from UTIs
- Line 182: capitalize Enterococcal
- Line 190: consider highlighting previous studies/data regarding the use of ampicillin/amoxicillin for resistant Enterococci strains in uncomplicated UTIs given superior urinary concentrations of these antimicrobial agents (excreted ~90% unchanged in urine)
- Lines 190-194: consider removing information pertaining to erythromycin for reasons stated prior
- Line 194: define VRE
- Line 196: capitalize Enterococcal; urinary tract infection previously defined
- Line 201: VRE will be previously defined in line 194
- Line 205: missing reference
- Line 208: missing punctuation
- Line 210: change “Enterococcus” to “Enterococci”

- Line 212: change “is” to “are”
- Line 213: capitalize Enterococci
- Lines 215-226: major grammatical errors in sentence structure (incomplete sentences)
 - Line 216: consider ending sentence after “> 50%”
 - Lines 218-219: “... acid, and increases the permeability of connective tissue which aids bacterial spread within tissues and is thought to be related to vancomycin resistance.”
 - Lines 220-221: change “is” to “are”; “... cells and are encoded by plasmids.”
 - Line 221: capitalize Enterococci
 - Lines 222-223: “clumping, facilitating conjugation, and mediating adhesion to host cells and extracellular matrixes.”
 - Line 226: Enterococci does not need to be italicized
- Line 230: rates, were
- Line 236: missing reference
- Line 242: remain consistent with percentage rounding to nearest tenth; belonging
- Line 245: urinary tract infections previously defined
- Line 246: “... to a previous article.”
- Line 248: sequence types previously defined in line 128
- Line 268: capitalize Enterococcal
- Line 273: capitalize Enterococci
- Lines 273-274: urinary tract infections previously defined
- Line 277: Enterococci does not need to be italicized
- Line 278: urinary tract infections previously defined
- Line 279: Enterococci does not need to be italicized
- Line 287: further elaborate on the random sampling methods utilized
- Line 294: capitalize Enterococcal
- Line 295: tigecycline or teicoplanin? If tigecycline was susceptibility tested, why were data not included in the study?
- Line 301: define DNA
- Line 309: MLST previously defined in line 128; *Enterococcus* should be italicized
- Line 312: change “*Enterococcus*” to “Enterococci”

Reviewer 1:

Comment 1: Introduction is dense with information, could it be simplified?

Response 1: We agree that the Introduction was initially dense. We have revised the Introduction section to simplify the content and enhance clarity, focusing on key points relevant to the study.

Comment 2: The Enterococcus clinical isolates were randomly selected from UTI patients, but the method was not mentioned in the text.

Response 2: Thank you for your insightful suggestion. We have now included a detailed explanation of the method used to randomly select the Enterococcus clinical isolates in the Methods section of the revised manuscript. Specifically, we utilized the random number generation function (RAND()) in Microsoft Excel to generate random numbers for each isolate, ensuring an unbiased selection process.

Comment 3: Table 4, Relationship of MLST phenotype with E. faecalis, antimicrobial resistance and virulence gene,

Do you mean "Relationship of MLST phenotype with antimicrobial resistance and virulence gene in E. faecalis" ?

Response 3: Thank you! You are correct in your observation. The title of Table 4 has been revised to "Relationship of MLST Phenotype with Antimicrobial Resistance and Virulence Genes in E. faecalis" to more accurately reflect the content of the table.

Comment 4: Table 4, It will be better to add proportion in brackets for antimicrobial resistance and virulence genes. For example, ST16, AMP,1 (3.7%)

Response 4: We appreciate this suggestion and have now added the

proportions in brackets for antimicrobial resistance and virulence genes in Table 4. For example, ST16, AMP, 1 (3.7%). This addition provides a clearer understanding of the data distribution.

Comment 5: Table 5, suggestions are similar to those for Table 4.

Response 5: Thank you! We have made similar updates to Table 5 to include proportions in brackets for antimicrobial resistance and virulence genes.

Comment 6: Line 237, ST4 *E. faecalis*, as there also exist *E. faecium*

Response 6: Thank you! We acknowledge the potential confusion. We have revised the manuscript to ensure a clear distinction between *E. faecalis* and *E. faecium*. The text now accurately reflects the correct species.

Comment 7: What is the mechanism of five *E. faecalis* isolates which were not susceptible to Linezolid? Whole genome sequencing is suggested to perform to get some clues.

Response 7: Thank you for your valuable suggestion. We agree that whole-genome sequencing (WGS) is essential for uncovering the mechanism of Linezolid resistance in these *E. faecalis* isolates. We have now conducted WGS on the five Linezolid-resistant isolates and have incorporated the findings into the manuscript. The sequencing data have provided important insights into the potential genetic mechanisms underlying the resistance. We have updated the Methods and Results sections to include these new data, and we have expanded the Discussion to analyze and interpret the WGS results.

Reviewer 2:

Comment 1: For the Antimicrobial Susceptibility Testing, disc diffusion method was used. The MICs of antimicrobials agents were not tested. For *Enterococcus* spp., fosfomycin is an important antibiotics and the study is not included. The antimicrobial susceptibility of teicoplanin was shown in the Results but not in the Method.

Response 1: We have now conducted MIC testing for the key antibiotics included in our study, in addition to the disc diffusion method initially used. This data has been incorporated into the Results section, providing a more comprehensive assessment of the antimicrobial susceptibility of the *Enterococcus* spp. Acknowledging the importance of fosfomycin as an effective treatment option for *Enterococcus* spp., we have now included fosfomycin in our antimicrobial susceptibility testing panel. The results of these tests have been added to the Results section. Thank you for pointing out this inconsistency. The omission of teicoplanin in the Methods section was indeed due to a typographical error. We apologize for any confusion this may have caused. The antimicrobial susceptibility testing for teicoplanin was conducted as part of our study, and we have now corrected the Methods section to accurately reflect this.

Comment 2: For the Detection of Virulence Factors, the positive amplified products should be sequenced to further confirm.

Response 2: Thank you for your valuable suggestion. We agree that sequencing the positive amplified products would provide additional confirmation of our results. We have now performed sequencing on these products to validate the presence of the virulence factors. We have included a statement in the Methods section indicating that the positive amplified products were sequenced for confirmation.

Comment 3: For the MLST typing, the primers of the housekeeping genes used in the manuscript should be showed at least in the supplementary materials.

Response 3: We appreciate your suggestion to include the primers used for the housekeeping genes in the MLST typing. We have now added these primer sequences to the supplementary materials, ensuring that all necessary details are available for replication and further study.

Comment 4: The part of introduction is too rigmorole.

Response 4: We agree that the Introduction was overly detailed. We have revised this section to streamline the content, focusing on the most relevant background information that directly supports the study's objectives. This should make the Introduction more concise and accessible.

Comment 5: The authors should distinguish the genes and proteins. For example, esp is gene and Esp is protein. The gene should be italic.

Response 5: Thank you for pointing out the distinction between genes and proteins. We have ensured that all gene names are italicized and that the distinction between genes and proteins is clearly made throughout the manuscript.

Comment 6: The manuscript is only a description draft and no mechanisms are studied. Too many tables are included, most of which are not meaningful.

Response 6: Thank you for your valuable suggestion. To address your concern, we have now conducted whole-genome sequencing (WGS) on the Linezolid-resistant isolates to investigate the underlying mechanisms of resistance. The sequencing data have provided important insights into this resistance. We have updated the Methods and Results sections to include these new findings and have expanded the Discussion to analyze and interpret

the WGS results.

Reviewer 3:

Comment 1: In the manuscript, the authors state that this study involves patient enrollment. According to current ethical requirements, such research usually requires the approval of an ethics committee. However, there is no information on whether or not ethical approval has been obtained in the manuscript.

Response 1: We acknowledge the need for ethical approval in studies involving patient enrollment. We apologize for the oversight and have now included a statement regarding the ethical approval obtained for this study. The study was reviewed and approved by the Ethics Committee of Shanghai Sixth People's Hospital (No.2024-KY-051K), and this information has been added to the Methods section of the manuscript.

Comment 2: The methods and judgment criteria for antimicrobial susceptibility testing should be listed. According to CLSI's routine recommendation, the quality control strain for KB method is *Staphylococcus aureus* ATCC25923, rather than *Enterococcus faecalis* ATCC29212.

Response 2: Thank you for pointing out this important detail. We have included detailed methods and judgment criteria for antimicrobial susceptibility testing in the revised Methods section, including the quality control strain used.

Reviewer 4

General suggestions

Comment 1. Include more clinical details and endpoints to baseline characteristics.

- o Were UTIs polymicrobial or monomicrobial?
- o Infection severity/type – cystitis, pyelonephritis, catheter-associated UTI?
- o Community or hospital-acquired infection?

Response 1: Thank you so much! We think this is an excellent suggestion. We have expanded the clinical details and baseline characteristics, including information on whether UTIs were polymicrobial or monomicrobial, infection severity/type, and whether infections were community or hospital-acquired in Table 1.

Comment 2. Include additional baseline characteristics regarding risk factors associated with Enterococcal UTIs as alluded to in lines 54-56.

- o Malignancy?
- o Steroid use or other immunosuppressing medications/conditions?
- o Prior antimicrobial use?

Response 2: Thank you so much! We have added additional baseline characteristics related to risk factors such as malignancy, steroid use, and prior antimicrobial use in Table 1.

Comment 3. Address study limitations in the discussion.

Response 3: Thank you for your remind! We have included a section addressing the study limitations in the Discussion to provide a balanced view of the findings.

Comment 4. Discussion in general is lengthy and lacks logical flow; would suggest refining (e.g., consider moving gene descriptions to the introduction)

Response 4: Thank you so much! We have refined the Discussion section for better logical flow and clarity, and we have moved gene descriptions to the Introduction as suggested.

Detailed suggestions

- Line 19: add “select” in front of Enterococcal strains since only species *faecalis* and *faecium* were observed; Enterococcal does not need to be italicized

R: Thank you for your valuable suggestion! We have added “select” in front of “Enterococcal strains” and corrected the formatting of “Enterococcal” in the revised manuscript.

- Line 20: spell out numbers (e.g., eighty rather than 80)

R: Thank you for your suggestion! We have revised the manuscript to spell out the numbers as recommended.

- Line 23: define PCR before utilizing abbreviation

R: Thank you for pointing this out! We have now defined “PCR” before using the abbreviation in the revised manuscript.

- Lines 40-41: sentence tense changes; suggest separating into two separate sentences

R: Thank you! We've adjusted it in our revised manuscript.

- Line 41: missing reference

R: Thank you for highlighting this! We've adjusted it in our revised manuscript.

- Line 43: restructure sentence to, “ ... 20.7% of pathogens were isolated from urine cultures ... ”

R: Thank you! We've adjusted it in our revised manuscript.

- Line 47: UTI was already previously defined in line 37

R: Thank you for your reminder! We have removed the redundant definition of “UTI” in the revised manuscript.

- Line 52: Enterococci emerge

R: Thank you! We've adjusted it in our revised manuscript.

- Lines 56-57: Enterococci contribute

R: Thank you! We've adjusted it in our revised manuscript.

- Lines 65, 67, 69, 71, 74: add “species” following *Enterococcus*

R: Thank you for your careful review! We have added “species” following “Enterococcus” as suggested.

- Line 70: missing reference

R: Thank you for your remind! We've added the reference in our revised manuscript.

- Lines 73-74: add “select” in front of *Enterococcus*

R: Thank you for bringing this to our attention! We have added the missing reference in the revised manuscript.

- Line 75: Enterococci does not need to be italicized; since all UTIs in this cohort were not hospital-acquired, it may not be appropriate to conclude that the findings of this study will inform management of nosocomial infections; you may consider a subgroup analysis describing antimicrobial resistance patterns in patients from a hospitalized origin versus patients from a community origin to see if there are any notable differences to highlight.

R: Thank you for your insightful suggestion! We have conducted the subgroup analysis as recommended, comparing antimicrobial resistance patterns between patients from hospitalized origins and those from community origins. For *Enterococcus faecium*, we did not perform subgroup analysis due to the limited number of isolates from community origins. However, for *Enterococcus faecalis*, the subgroup analysis did not reveal any significant differences between the two groups. The results of the subgroup analysis for *Enterococcus faecalis* are summarized in the table below for your reference. We have chosen not to include this table in the final manuscript but are happy to provide it here to illustrate our findings.

Table: Subgroup analysis of antimicrobial resistance patterns in *Enterococcus faecalis* between hospitalized and community-Origin Patients

Antibiotics	Hospitalized (n=65)	Community (n=15)	P
% Resistant Isolates			
Ampicillin	6.2	6.7	0.941
Gentamicin	40	13.3	0.051
Nitrofurantoin	1.5	6.7	0.252
Fosfomycin	3.1	6.7	0.509
Linezolid	6.2	6.7	0.941
Vancomycin	0.0	0.0	-
Teicoplanin	0.0	0.0	-

- Line 81: remove "s" from UTIs

R: Thank you for your reminder! We have removed the "s" from "UTIs" in the revised manuscript.

- Lines 81-82: spell out "years" rather than using "y"

R: Thank you! We have revised this section and spelled out "years" as suggested.

- Lines 83-84: remain consistent with percentage rounding to nearest tenth as in above lines

R: Thank you for this important note! We have ensured consistency with percentage rounding throughout the manuscript.

- Table 1

- o Define Efa and Efm as footnote below table or next to the respective species

R:Thank you! We've added the footnote below table.

- o Consider adding other baseline characteristics that place patients at increased risk for Enterococcal UTIs as alluded to in lines 54-57 (e.g., severity of illness, previous antimicrobial use, malignancy, immunosuppressive medications, UTI type)

R:Thank you for professional comments. We have adding other characteristics such as infection type (community, hospital-acquired infection, catheter-associated, polymicrobial, polymicrobial), drug usage

(steroid/Immunosuppressor and antibiotics) and comorbidities (urological malignancy, urinary calculi and pneumonia) in Table 1.

- Line 90: linezolid does not need to be capitalized

R:Thank you for your remind! We've adjusted it in our revised manuscript.

- Lines 90-93: consider removing the statement and data for erythromycin; due to

increasing in vivo resistance and poor urinary penetration of macrolide antimicrobials (excreted 6-14% unchanged in the urine), they are not reliable options clinically for the treatment of Enterococcal UTI

R:Thank you for your remind! We've removed the statement and data for erythromycin in our revised manuscript.

- Lines 100-102: correct "tigecycline" to "teicoplanin"; remove references to Chi-square value since this statistic was not displayed in table 2

R:Thank you! We've corrected "tigecycline" to "teicoplanin" and removed references to Chi-square value.

- Lines 130-131: remain consistent with percentage rounding to nearest tenth as in above lines

R:Thank you for your remind! We've adjusted it in our revised manuscript.

- Line 135: remain consistent with order of data presented (n, %)

R:Thank you for your remind! We've adjusted it in our revised manuscript.

- Line 143: remove "s" from UTIs

R:Thank you! We've removed "s" from UTIs.

- Table 4: define the antimicrobial abbreviations used as a footnote (e.g., AMP = ampicillin)

R:Thank you for your remind! We've define the antimicrobial abbreviations used as a footnote.

(Note:AMP=ampicillin,GEH=gentamicin,NIT=nitrofurantoin,FOS=fosfomycin,LNZ=linezolidTEC=teicoplanin,VAN=vancomycin.)

- Line 164: Enterococci does not need to be italicized

R:Thank you for your remind! We've adjusted it in our revised manuscript.

- Line 166: missing reference

R:Thank you for your remind! We've added the reference in our revised manuscript.

- Line 167: UTIs already defined; missing reference

R:Thank you! We've adjusted it and added the reference in our revised manuscript.

- Line 171: capitalize Enterococci

R:Thank you! We have corrected it.

- Line 175: capitalize Enterococci; urinary tract infection already defined

R:Thank you! We have corrected it in italics and adjusted “urinary tract infection”.

- Lines 176-177: sentence does not make grammatical sense; consider restructuring as follows: “UTIs affect diverse populations with a higher incidence rate among females, ranging from 62.6% to 86.9%.”

R:Thank you for your remind! We've adjusted it in our revised manuscript.

- Line 178: capitalize Enterococci

R:Thank you! We've adjusted it in our revised manuscript.

- Line 179: remove “s” from UTIs

R:Thank you! We have removed “s” from UTIs.

- Line 182: capitalize Enterococcal

R:Thank you! We've adjusted it in our revised manuscript.

- Line 190: consider highlighting previous studies/data regarding the use of

ampicillin/amoxicillin for resistant Enterococci strains in uncomplicated UTIs given superior urinary concentrations of these antimicrobial agents (excreted ~90% unchanged in urine)

R:Thank you for professional comments.We've highlighted previous studies regarding the use of ampicillin for resistant Enterococci strains in UTIs.

- Lines 190-194: consider removing information pertaining to erythromycin for reasons stated prior

R:Thank you! We've removed information pertaining to erythromycin in our revised manuscript.

- Line 194: define VRE

R:Thank you! We've adjusted it in our revised manuscript.

- Line 196: capitalize Enterococcal; urinary tract infection previously defined

R:Thank you! We've adjusted it in our revised manuscript.

- Line 201: VRE will be previously defined in line 194

R:Thank you! We've adjusted it in our revised manuscript.

- Line 205: missing reference

R:Thank you for your remind! We've added the reference in our revised manuscript.

- Line 208: missing punctuation

R:Thank you! We've adjusted it in our revised manuscript.

- Line 210: change “Enterococcus” to “Enterococci”

R:Thank you! We've adjusted it in our revised manuscript.

- Line 212: change “is” to “are”

R:Thank you! We've adjusted it in our revised manuscript.

- Line 213: capitalize Enterococci

R:Thank you! We've adjusted it in our revised manuscript.

- Lines 215-226: major grammatical errors in sentence structure (incomplete sentences)
 - o Line 216: consider ending sentence after "> 50%"
R:Thank you! We've adjusted it in our revised manuscript.
 - o Lines 218-219: "... acid, and increases the permeability of connective tissue which aids bacterial spread within tissues and is thought to be related to vancomycin resistance."
R:Thank you! We've adjusted it in our revised manuscript.
 - o Lines 220-221: change "is" to "are"; "... cells and are encoded by plasmids."
R:Thank you! We've adjusted it in our revised manuscript.
 - o Line 221: capitalize Enterococci
R:Thank you! We've adjusted it in our revised manuscript.
 - o Lines 222-223: "clumping, facilitating conjugation, and mediating adhesion to host cells and extracellular matrixes."
R:Thank you! We've adjusted it in our revised manuscript.
 - o Line 226: Enterococci does not need to be italicized
R:Thank you! We've adjusted it in our revised manuscript.
- Line 230: rates, were
R:Thank you! We've adjusted it in our revised manuscript.
- Line 236: missing reference
R:Thank you for your remind! We've added the reference in our revised manuscript.
- Line 242: remain consistent with percentage rounding to nearest tenth; belonging
R:Thank you! We've adjusted it in our revised manuscript.
- Line 245: urinary tract infections previously defined
R:Thank you! We've adjusted it in our revised manuscript.
- Line 246: "... to a previous article."
R:Thank you! We've adjusted it in our revised manuscript.
- Line 248: sequence types previously defined in line 128
R:Thank you! We've adjusted it in our revised manuscript.
- Line 268: capitalize Enterococcal
R:Thank you for your remind! We've capitalize Enterococcal.
- Line 273: capitalize Enterococci
R:Thank you for your remind! We've capitalize Enterococci.
- Lines 273-274: urinary tract infections previously defined
R:Thank you! We've adjusted it in our revised manuscript.
- Line 277: Enterococci does not need to be italicized
R:Thank you! We've adjusted it in our revised manuscript.
- Line 278: urinary tract infections previously defined
R:Thank you! We've adjusted it in our revised manuscript.
- Line 279: Enterococci does not need to be italicized
R:Thank you! We've adjusted it in our revised manuscript.
- Line 287: further elaborate on the random sampling methods utilized

R:Thank you! We've utilized further elaborate on the random sampling methods.

- Line 294: capitalize Enterococcal

R:Thank you! We've capitalized Enterococcal.

- Line 295: tigecycline or teicoplanin? If tigecycline was susceptibility tested, why were data not included in the study?

R:We were really sorry for our careless mistakes. Thank you for your remind!This should be teicoplanin.

- Line 301: define DNA

R:Thank you! We've defined DNA.

- Line 309: MLST previously defined in line 128; *Enterococcus* should be italicized

R:Thank you! We've adjusted it in our revised manuscript.

- Line 312: change "*Enterococcus*" to "Enterococci"

R:Thank you! We've adjusted it in our revised manuscript.

Re: Spectrum01217-24R1 (Antimicrobial Resistance, Virulence Gene Profiles, and Molecular Epidemiology of Enterococcal Isolates from Patients with Urinary Tract Infections in Shanghai, China)

Dear Ms. Qiong Wu:

Thank you for the privilege of reviewing your work. Below you will find my comments, instructions from the Spectrum editorial office, and the reviewer comments.

Revision Guidelines

Sincerely,
Tomefa Asempa
Editor
Microbiology Spectrum

Reviewer #1 (Comments for the Author):

The authors have addressed all my concerns

Reviewer #2 (Comments for the Author):

The manuscript written by Weiyi Wu et al has been improved after revision. Some minor comments are as follows.

1. The table 2 and table 3 could be merged.
2. Table 4: the genes should be italic.
3. Table 5: the P value of gelE is 0? Please confirm it.

The manuscript written by Weiyi Wu et al has been improved after revision. Some minor comments are as follows.

1. The table 2 and table 3 could be merged.
2. Table 4: the genes should be italic.
3. Table 5: the P value of gelE is 0? Please confirm it.

Reviewer 1:

Comment 1: The authors have addressed all my concerns

Response 1: Thank you very much for your positive feedback and for taking the time to review our manuscript. We greatly appreciate your valuable comments and suggestions, which have helped us improve the quality of the paper.

Reviewer 2:

Comment 1: The table 2 and table 3 could be merged.

Response 1: Thank you for your valuable suggestion. We agree with your suggestion and have merged Table 2 and Table 3 into a single table for better clarity and conciseness. The revised table is now presented as Table 2.

Comment 2: Table 4: the genes should be italic.

Response 2: Thank you for pointing this out. We have corrected the formatting of the gene names, and all gene names are now italicized as per the standard convention.

Comment 3: Table 5: the P value of *gelE* is 0? Please confirm it.

Response 3: Thank you for your inquiry regarding Table 5. After a thorough review, We can confirm that the P value for *gelE* is 0.000. This representation provides a more accurate reflection of the statistical significance, and we appreciate your attention to this detail.

Re: Spectrum01217-24R2 (Antimicrobial Resistance, Virulence Gene Profiles, and Molecular Epidemiology of Enterococcal Isolates from Patients with Urinary Tract Infections in Shanghai, China)

Dear Ms. Qiong Wu:

Thank you for the privilege of reviewing your work. Below you will find my comments, instructions from the Spectrum editorial office.

Line 41-42. please revise as sentence grammar is confusing.

Line 72: please revise grammar "verified attribute"

Confirm p value of Antibiotics in Table 1.

Line 174: harbored spelling

Please return the manuscript within 10 days; if you cannot complete the modification within this time period, please contact me. If you do not wish to modify the manuscript and prefer to submit it to another journal, notify me immediately so that the manuscript may be formally withdrawn from consideration by Spectrum.

Revision Guidelines

Sincerely,
Tomefa Asempa
Editor
Microbiology Spectrum

Spectrum editorial office

Comment 1: Line 41-42. please revise as sentence grammar is confusing.

Response 1: Thank you for pointing this out. We have revised the sentence to improve clarity and correct the grammatical issues. The revised sentence now reads, "Linezolid-resistant strains, all carrying the *optrA* gene, were identified, including the first report of ST300 in China."

Comment 2: Line 72: please revise grammar "verified attribute"

Response 2: Thank you! We have reviewed and revised the phrase to correct the grammar. The revised sentence now reads, "Resistance to linezolid was attributed to mutations in the 23S rRNA and genes encoding the 50S ribosomal proteins L3 (*rpIC*) and L4(*rpID*), the acquisition of genes such as *optrA* and *poxtA* encoding an ATP-binding cassette (ABC-F) protein and *cfr* variants coding for 23S rRNA methyltransferase."

Comment 3: Confirm p value of Antibiotics in Table 1.

Response 3: Thank you for your inquiry regarding Table 1. After a thorough review, We can confirm that the P value for Antibiotics is 0.000. This representation provides a more accurate reflection of the statistical significance, and we appreciate your attention to this detail.

Comment 4: Line 174: harbored spelling

Response 4: Thank you for pointing this out. We apologize for the oversight. The spelling has now been corrected.

Re: Spectrum01217-24R3 (Antimicrobial Resistance, Virulence Gene Profiles, and Molecular Epidemiology of Enterococcal Isolates from Patients with Urinary Tract Infections in Shanghai, China)

Dear Ms. Qiong Wu:

Your manuscript has been accepted, and I am forwarding it to the ASM production staff for publication. Your paper will first be checked to make sure all elements meet the technical requirements. ASM staff will contact you if anything needs to be revised before copyediting and production can begin. Otherwise, you will be notified when your proofs are ready to be viewed.

Sincerely,
Tomefa Asempa
Editor
Microbiology Spectrum